# Stem Cell Theory of Cancer: Clinical Implications of Epigenomic versus Genomic Biomarkers in Cancer Care

**DOI:** 10.3390/cancers15235533

**Published:** 2023-11-22

**Authors:** Shi-Ming Tu, Jim Zhongning Chen, Sunny R. Singh, Ahmet Murat Aydin, Neriman Gokden, Neville Ngai Chung Tam, Yuet-Kin Leung, Timothy Langford, Shuk-Mei Ho

**Affiliations:** 1Division of Hematology and Oncology, University of Arkansas for Medical Sciences, Little Rock, AR 72205, USA; zchen3@uams.edu (J.Z.C.); srsingh@uams.edu (S.R.S.); 2Department of Urology, University of Arkansas for Medical Sciences, Little Rock, AR 72205, USA; maydin@uams.edu (A.M.A.); tlangford@uams.edu (T.L.); 3Department of Pathology, University of Arkansas for Medical Sciences, Little Rock, AR 72205, USA; gokdenneriman@uams.edu; 4Department of Pharmacology and Toxicology, University of Arkansas for Medical Sciences, Little Rock, AR 72205, USA; nevilletam@uams.edu (N.N.C.T.); rickyleung@uams.edu (Y.-K.L.); shukmeiho@uams.edu (S.-M.H.)

**Keywords:** biomarkers, cancer stem cells, heterogeneity, epithelial–mesenchymal transition, circulating tumor cells, ctDNA, epigenomic

## Abstract

**Simple Summary:**

A stem cell origin and nature of a cancer suggests that genetic biomarkers may be pivotal, but cellular context is paramount. When we consider cellular context and when it concerns separation of cancer stem cells (CSCs) from non-CSCs, whether the biomarker is derived from the DNA, RNA, or protein and whether it is detected within the tumor or in the blood (such as CTCs and ctDNA) has biological ramifications and clinical implications for cancer care. Importantly, if CSCs are a relevant biological and clinical entity, then measuring, monitoring, and modulating via genome versus epigenome become imperative for diagnosis, prognosis, and therapeutics in cancer care.

**Abstract:**

Biomarkers play a crucial role in the diagnosis, prognosis, and therapeutics of cancer. We use biomarkers to identify, image, monitor, and target cancer. In many respects, the discovery of pertinent biomarkers that distinguish fulminant from indolent neoplasms and sensitive from refractory malignancies would be a holy grail of cancer research and therapy. We propose that a stem cell versus genetic theory of cancer may not only enable us to track and trace the biological evolution of cancer but also empower us to attenuate its clinical course and optimize the clinical outcome of patients with cancer. Hence, a biomarker that identifies cancer stem cells (CSCs) and distinguishes them from non-CSCs may serve to elucidate inter-tumoral and intra-tumoral heterogeneity, elevate the values and utility of current prognostic and predictive tests, and enhance drug versus therapy development in cancer care. From this perspective, we focus on CSC biomarkers and discuss stemness or stem-like biomarkers in the context of a unified theory and a consideration of stem cell versus genetic origin. We review their role in primary and mixed tumors, in the elaboration of tumor subtypes, and in the imaging and monitoring of minimal residual diseases. We investigate how scientific theories influence the direction of scientific research and interpretation of experimental results, and how genomics and epigenomics affect the dynamics and trajectories of biomarkers in the conduct of cancer research and in the practice of cancer care.


*A good decision is based on knowledge not numbers.*
Plato

## 1. Introduction

Biomarkers can be instrumental in the diagnosis, prognosis, and therapeutics of cancer. We use biomarkers to identify, measure, image, and target cancer. But without good knowledge and clear understanding about the origin and nature of cancer, the data we collect may not be relevant, the numbers we compile may not be applicable, and the decision we make may not be appropriate for cancer care.

In many respects, the discovery of pertinent biomarkers that distinguish fulminant from indolent neoplasms and responsive from refractory malignancies would be a holy grail of cancer research and therapy.

However, searching for a germane but elusive cancer biomarker could be quixotic rather than rewarding without a proper cancer theory. We will be seeking a proverbial biomarker needle in the cancer haystack. We may discover biomarkers with less or no prognostic and predictive value because they represent passengers rather than drivers, correlations rather than causations, and mere markers rather than real makers of cancer.

We propose that a stem cell rather than genetic theory of cancer may not only enable us to trace and track the natural history of cancer but also empower us to attenuate its clinical course and optimize the clinical outcome for patients with cancer.

Hence, a biomarker that identifies cancer stem cells (0s) and distinguishes it from non-CSCs may serve to elucidate inter-tumoral and intra-tumoral heterogeneity, elevate the merits and values of current prognostic and predictive tests, and enhance ‘drug versus therapy’ development in cancer care (Figure 1).

From this perspective, we focus on CSC biomarkers and discuss stemness or stem-like biomarkers in the context of a consideration of a stem cell versus a genetic origin of cancer. We review their role in primary and mixed tumors, in the elaboration of tumor subtypes, and in the imaging and monitoring of minimal residual diseases. We investigate how scientific theories influence the collection and interpretation of experimental results, and how genomics and epigenomics affect the dynamics and trajectories of biomarkers in the conduct of cancer research and in the practice of cancer care.

## 2. Brief History

In 1847, Bence Jones first reported a biomarker in the urine of a patient with multiple myeloma [1,2]. Today, we still use this biomarker, a monoclonal light chain immunoglobulin protein, to diagnose and monitor multiple myeloma.

In the 1960s, introduction of immunological techniques such as radioimmunoassay facilitated the discovery of carcinoembryonic antigen (CEA) and alfa-fetoprotein (AFP) [3,4].

In the 1980s, advent of hybridoma technology ushered the development of prostate-specific antigen (PSA) and carbohydrate antigen (CA) 125 [5,6].

In many respects, PSA is a paragon cancer biomarker: specific, sensitive, reliable, and quantifiable. Unfortunately, it is a perfect biomarker in one sense, but also suffers from an inherent imperfection. Although it can reveal a large part of an overt tumor, it cannot uncover the hidden source of the whole tumor. In other words, it detects non-CSCs but misses CSCs.

Although technology revolutionizes science, we propose that knowledge as encapsulated in a proper cancer theory could upgrade both, and elevate the relevance of cancer research and the quality and safety of cancer care for the near future, as in the recent past [7,8].

## 3. Origin of Cancer

According to the scientific method, a pertinent scientific theory based on credible observations empowers scientific research [9]. It enables the appropriate design of experiments to test relevant hypotheses pertaining to indisputable observations.

Perhaps a pertinent theory is in the mind of the believer and a credible observation is in the eyes of the beholder. A potential problem with the scientific method is that it does not preclude us from performing exemplary experiments to test an erroneous theory based on faulty observations (such as those generated in the laboratory rather than in real life, in which the experiments were conducted to generate rather than to test the hypotheses).

For example, according to a genetic theory of cancer and a multistep model of carcinogenesis, subclones with additional genetic defects become metastatic and have more malignant properties [10]. Biomarkers based on these genetic aberrations are useful for the diagnosis, prognosis, and therapeutics of diverse cancers. But, overall, the clinical impact tends to be underwhelming—marginal, modest, and momentary [11]. We try to fix the theory to fit the data by resorting to truncal versus branch mutations [12]. Perhaps the biomarkers are alluding to another insight, above genomics and concerning epigenomics, i.e., that we may be dealing with different tumor subtypes and phenotypes.

In contrast, a stem cell theory of cancer predicts that malignancies derived from earlier progenitor stem cells have increased malignant potential, such as metastatic tendency, intra-tumoral heterogeneity, drug resistance, immune evasion, etc., regardless of genetic mutations, many of which are rare rather than regular in nature. Contrary to the genetic theory of cancer, more advanced and aggressive cancers may have fewer mutations than their less advanced and more indolent counterparts [7,8]. Perhaps the truncal and branch mutations help us to distinguish progenitor CSCs from progeny non-CSCs. But if cellular context trumps genetic content, then it is conceivable that biomarkers that distinguish CSCs from non-CSCs have unfulfilled scientific prospects in cancer research and untapped clinical potential in cancer care.

Nowadays, there is controversy about a stem cell theory of cancer [13,14,15]. A stem cell theory may provide us with a comprehensive understanding and knowledge of the origin and nature of cancer [7,8]. It may be the elusive unified theory of cancer that elucidates the origin of all cancer hallmarks, including heterogeneity, metastasis, and drug resistance [7,8]. It embraces the genomics and epigenomics of cancer. It unites various compartments, different components, and the microenvironment of cancer.

Kamel et al. [2] highlighted a deliberate design and a systemic approach to identify relevant cancer biomarkers. However, it is important to remind ourselves that, when we do not adopt or adhere to the basic principles of the scientific method, there is a chance that what we discover may be less useful, if not useless, for cancer care. When we happen to find some scientific merits and clinical values in some biomarkers in this way, we may have done so for the wrong reasons.

## 4. Diagnostic/Prognostic Biomarkers

### 4.1. Primary Tumors

A biomarker that forecasts the future behavior of a primary tumor (such as metastatic potential, which affects the overall survival of the patient) is a powerful biomarker indeed. Such power is likely to be vested in CSCs rather than non-CSCs, because the former are more likely than the latter to regenerate, propagate, and disseminate.

Therefore, a CSC biomarker may serve a different purpose and impart a different value compared with a non-CSC biomarker in a primary tumor; the power of prognostication belongs to the former.

Often enough, the primary tumor is where we obtain tumor tissue from which we establish the diagnosis of cancer. A biopsy of the primary tumor provides us with the initial diagnosis, overall prognosis, and tentative treatment plans for the patient.

There is interest in detecting biomarkers in a primary tumor that provide diagnostic, prognostic, and predictive values for the entire tumor. In many respects, these clinical parameters are closely interrelated. A biomarker that establishes the diagnosis of a particular subtype of cancer, empowering the prognostication of its indolence versus virulence and enabling the design of breakthrough versus marginal treatments, is a dream for all cancer researchers and practitioners.

For example, small cell cancer (SCLC) is distinct from non-small cell cancer (NSCLC) of the lung, and triple-negative breast cancer (TNBC) is different from estrogen/progesterone receptor positive (HR+) breast cancer. However, there are instances where a NSCLC may be endowed with SCLC properties and HR+ breast cancer has the power to transform into TNBC-like breast cancer (i.e., it may be a mixed tumor). Is this truly transformation, or is it revelation of its real identity? Could we identify such a treacherous tumor at its nascent stage in a primary tumor?

If a primary tumor with elevated metastatic potential is likely to be threatening, then detecting metastatic biomarkers in primary tumors would be informative and useful, except that metastasis is supposed to develop later rather than earlier (according to Vogelstein’s model of multistep carcinogenesis) and, if it is present in the primary tumor, it contradicts the precise definition and is contrary to the very premise of an authentic metastatic biomarker.

Suppose there are malignant subclones embedded in the primary tumor that have truncal mutations and exhibit unique genetic (e.g., metastatic) signatures or epigenetic (e.g., stemness) profiles. Perhaps these malignant clones from single-cell sequencing will display epithelial-to-mesenchymal transition (EMT) expression, embryonic signaling, and/or inflammatory activity. Although putative malignant subclones are sporadic and elusive in a primary tumor [16,17], perhaps we can still find them by means of spatial transcriptomics or proteomics at the level of single-cell resolution [18,19].

According to a stem cell origin of cancer, we may have a problem with the discovery of pertinent stemness/stem-like biomarkers in a primary tumor. Shin et al. [20] tested the hypothesis of a cellular origin of bladder cancer in a chemical carcinogenesis model. Without prior bias regarding genetic pathways or cell types, they prospectively marked or ablated cells to demonstrate that muscle-invasive bladder carcinomas exclusively arose from Shh-expressing stem cells in the basal urothelium. These carcinomas originated from a single cell whose progeny aggressively colonized a major portion of the urothelium to generate a lesion with histological features identical to human carcinoma in situ. Shh-expressing basal cells within this precursor lesion were the tumor-initiating cells, although Shh expression itself was lost in subsequent carcinomas. Therefore, a subsequent “clinical” tumor phenotype could diverge significantly from that of the initial primeval cancer cell of origin.

### 4.2. Mixed Tumors

Intra-tumoral heterogeneity in a mixed tumor poses a different challenge to the utility of cancer biomarkers. Because pluripotent progenitor stem cells have the capacity to differentiate into multiple lineages, it is plausible that a CSC biomarker which reflects a progenitor stem cell may be able to capture all the diversity and variety in a mixed tumor that a non-CSC biomarker, which represents a progeny differentiated cell, may not [21].

A mixed tumor is a blatant example of an intra-tumoral heterogeneity that defies the simplicity of targeted therapy and challenges the fallacy of precision medicine.

#### 4.2.1. Testicular Cancer

About 80% of non-seminomatous germ cell tumors of the testis (NSGCT) are mixed tumors comprising various proportions of embryonal carcinoma, choriocarcinoma, yolk sac tumor, seminoma, and/or teratoma. Due to their common clonal origin, these vastly different and distinct tumors with diverse biological and clinical phenotypes have a similar if not identical genetic makeup [22]. In other words, a completely chemo-sensitive embryonal carcinoma that tends to be widely metastatic will have i(12p) and the same molecular profile as a completely chemo-resistant teratoma that tends to be indolent in the same mixed tumor.

Understandably, to cure a mixed NSGCT that contains embryonal carcinoma and teratoma, we resort to multimodal therapy rather than targeted therapy, which comprises integrated medicine rather than precision medicine [23]; we use systemic treatment (such as chemotherapy) to eradicate embryonal carcinoma, and local therapy (such as surgery) to extirpate teratoma. Anything less may provide modest, marginal, and momentary incremental clinical benefit at best, but is unlikely to lead to a curative clinical outcome.

#### 4.2.2. Breast Cancer

Breast cancer is divided into invasive ductal (80%) and lobular (about 15%) subgroups. However, a mixed ductal-lobular (MDL) subgroup (up to 5%) has been recognized (defined as a ductal component constituting at least 10% and a lobular component at least 50% of the tumor).

McCart-Reed et al. [24] published results that support a model in which separate morphological components of MDLs originate from a common ancestor, and that the lobular morphology can arise via a ductal pathway of tumor progression.

#### 4.2.3. Lung Cancer

Similarly, about 10% of those patients with small cell lung cancer (SCLC) may in fact have combined or mixed SCLC and non-small cell lung cancer (NSCLC).

Patients with mixed SCLC and NSCLC have a decreased overall survival (OS) compared with those with pure SCLC. Approximately 75% of the identified somatic mutations were present in both components. The findings suggest a common precursor with subsequent acquisition of oncogenic changes in mixed lung cancer [25].

#### 4.2.4. Osteosarcoma

Rajan et al. [26] showed that osteosarcomas displayed a high degree of cell–cell homogeneity with little sub-clonal diversification, despite extensive structural complexity. Phylogenetic analysis suggested that a majority of SCNAs were acquired during the early oncogenic process with relatively few structure-altering events arising in response to therapy or adaptation to growth in metastatic tissues. 

#### 4.2.5. Additional Solid Tumors

Again, malignant mixed tumors, as in the case of carcinosarcomas, present a unique experiment of nature in which we have an opportunity to elucidate the origin of cancer. Hence, the findings of a monoclonal origin of carcinosarcomas (including gliosarcomas) support a single pluripotential stem cell origin of cancer [27,28,29].

Finally, in tumors with concurrent urothelial, squamous, sarcomatoid, and glandular components, identical FISH abnormalities were noted in these separate components [30].

### 4.3. Tumor Subtypes

One way to solve any dilemma in the diagnosis, prognosis, and therapeutics of primary tumors or mixed tumors is to use biomarkers that categorize a whole array of disparate tumors into distinct subgroups based on their biological and clinical phenotypes.

Tumor subtyping may overcome the challenges of inter-tumoral and intra-tumoral heterogeneity, as well as offer a unique opportunity to advance the utility of cancer biomarkers. Because distinct progenitor stem cells lower in a stem cell hierarchy have a decreased capacity to express stemness properties, it is conceivable that a CSC biomarker (e.g., BAP1) will identify a tumor subtype (e.g., ccRCC4) that is clinically fulminant compared with a non-CSC biomarker (e.g., VEGFR), which will recognize another tumor subtype (e.g., ccRCC3) that is clinically indolent [31,32,33].

Nevertheless, the question remains at to which biomarkers based on which methods (e.g., genomics versus epigenomics) are more pertinent and practical for the purposes of subtyping tumors. There are pros and cons with either approach. We will illustrate potential advantages versus disadvantages of subtyping tumors based on their genomics versus epigenomics for the diagnosis, prognosis, and therapeutics of some selected tumor types.

#### 4.3.1. Breast Cancer

Breast cancer is a prototype tumor in which gene expression analysis first enabled its classification into distinct subgroups or intrinsic subtypes with clinical relevance: luminal A, luminal B, basal, and HER2 enriched [34]. There is substantial but imperfect correlation between classification according to the intrinsic subtypes and routine immunostaining for ER, PR, and HER2 in clinical practice.

Genomics. HER2 is an exemplary biomarker that fulfills the diagnosis, prognosis, and therapeutics of a specific subtype of breast cancer. Interestingly, HER2 may be linked to stemness and is believed to play a central role in the intrinsic regulatory pathways of breast cancer stem cells and their interactions with the tumor microenvironment [35].

Epigenomics. Triple-negative breast cancer (TNBC) belongs to the basal subtype. Trop2, a stem-like biomarker, expressed in 85% of TNBCs. It is also highly expressed in stem cells within various organs during embryogenesis [36]. Trop-2 signaling enhances the stem cell-like properties of cancer cells through beta-catenin signaling. Trop-2 targeting therapy comprising Sacituzumab-govitecan is approved for the treatment of refractory TNBC [37]. It is of interest that its related protein Epcam (Trop1) may also be a stem cell marker for the identification of circulating tumor cells (CTC) [38].

It is well known that chemotherapy is indispensable; it is the mainstay and treatment of choice for patients with fulminant cancers, such as TNBC (and SCLC, as well as embryonal carcinoma), in which there are no pertinent targetable or actionable mutations. Chemotherapy is also a pivotal treatment for patients with HER2+ tumors. Although targeted anti-HER2 therapy is beneficial, it is inadequate on its own without chemotherapy. We speculate that there should be a biological rationale, if not a logical reason, for combining anti-HER2 therapy with chemotherapy in the optimal treatment of HER2+ tumors.

#### 4.3.2. Prostate Cancer

Genomics. Although 75% of primary prostate cancer can be categorized into seven subtypes [39], one wonders whether the subdivisions are correct or convenient. For example, 59% of primary prostate cancers had ETS family gene fusions involving ERG, ETV1, ETV4, or FLI1. However, individuals with fusion-bearing tumors had a similar prognosis than those without following prostatectomy [40,41]. Furthermore, copy-number alterations and mutations were remarkably similar in their subtype distribution, and there was significant diversity in DNA copy-number alterations, gene expression, and DNA methylation within the subgroups.

Van Dessel et al. [42] found eight distinct genomic clusters in metastatic castration-resistant prostate cancer (CRPC). About 25% were actionable (7% with immunotherapy and 18% with poly-ADP-ribose polymerase inhibitor therapy). However, 66% did not harbor any clinically relevant or biologically evident genomics, suggesting an absence of actionable oncogenic drivers. Furthermore, the metastatic CRPC-derived genomic subgroups were not detectable in the primary prostate cancer cohort. Importantly, patients might have overlapped rather than distinctive genomics.

Epigenomics. Labrecque et al. [43] demonstrated five metastatic CRPC subtypes from their rapid autopsy study based on gene expression profiles. Their results suggest that metastatic CRPC is a disease continuum, i.e., some subtypes can convert to other subtypes.

Similarly, Tang et al. [44] discovered four CRPC subtypes derived from 40 PCa organoid and cell line models. They found no statistical difference in PTEN and TP53 alterations between the androgen receptor (AR)-dependent vs. independent subtypes. Although stem cell-like tumors had acquired lineage plasticity just like their neuroendocrine counterparts, they were driven by different master transcription factors, resulting in a different phenotype. As expected, stem cell-like tumors were less likely to respond to androgen receptor signaling inhibitor (ARSI) treatments.

#### 4.3.3. Bladder Cancer

Genomics. Choi et al. [45] showed that the basal subtype of bladder cancer displayed sarcomatoid, EMT, and stem cell-like features and was chemo-sensitive, while the luminal subtype contained FGFR3 and ERBB3 mutations. In addition, the luminal subtype could be further subdivided into luminal and p53-like and was chemo-resistant. However, mutant p53 was present at about the same frequency among the basal, luminal, and p53-like subtypes.

Robertson et al. [46] refined and extended this finding in 412 cases and 5 bladder cancer subtypes and managed to align biological phenotypes with therapeutic implications (Table 1).

Epigenomics. The Bladder Cancer Molecular Taxonomy Group [47] used 1750 MIBC transcriptomes and a network-based analysis of six independent MIBC classification systems to identify a consensus set of six molecular classes based on underlying oncogenic mechanisms, infiltration by immune stromal cells, and histological and clinical characteristics: luminal-papillary, luminal non-specified, luminal unstable, stroma-rich, basal/squamous, and neuroendocrine-like subtypes. Of note, TP53 mutations were found in 76%, 58%, and 94% of luminal unstable, basal/squamous, and neuroendocrine-like subtypes, respectively.

#### 4.3.4. Kidney Cancer

In an era when we have effective systemic treatments such as immunotherapy, comprising checkpoint inhibitors (CPIs) and tyrosine kinase inhibitors (TKIs). and evidence that certain patients benefit from local control of their disease (i.e., cytoreductive nephrectomy and metastasectomy), biomarkers may potentially guide us as to whom, what, when, and how to apply these treatment modalities in an effort to maximize any clinical benefit and optimize the clinical outcome in patient care [48,49].

Genomics. Turajlic et al. [50] demonstrated that certain patients tend to harbor more widespread occult metastases and would not benefit from surgery. He showed seven distinct evolutionary subtypes of renal cell carcinoma (RCC) based on intra-tumoral heterogeneity (ITH) and somatic copy-number alterations (SCNA) in primary kidney tumors. Results from their study suggest that removal of the primary tumor is beneficial for those patients with high ITH attenuated RCC, but not for those with low ITH and high SCNA fulminant RCC. For those patients with high SCNA, we postulate that their RCCs are likely to consist of aneuploidy, resulting from aberrant asymmetric division and derived from a progenitor cell with stemness properties (Table 2).

Epigenomics. Verbiest et al. [31] performed transcriptomics studies and confirmed prior findings in which RCC clear cell type (ccrcc) may be divided into four subtypes. Favorable subtypes (ccrcc 2—highly angiogenic, and ccrcc3—resembling normal kidney) repressed EMT and were associated with longer OS after diagnosis. In contrast, unfavorable subtypes (especially ccrcc4) demonstrated an aggressive stem cell-like phenotype with frequent sarcomatoid differentiation and inflammation-specific features at the pathologic level. Importantly, results from Verbiest’s epigenomic studies [31] confirm those from Turajlic’s genomic studies [50] and provide meaningful biological insights and feasible clinical utility (Table 3).

## 5. Predictive/Therapeutic Biomarkers

### 5.1. Monitoring

Although genetic markers can be very useful to trace and track the identity and evolution of cancer, they may be less useful to distinguish CSCs from non-CSCs, because both CSCs and non-CSCs within the same tumor share a similar, if not identical, genetic signature, due to their common clonal origin. Furthermore, if CSCs represent the root or trunk and non-CSCs the branches and leaves of cancer, then genetic testing may miss the former due to its paucity and obscurity within the whole tumor.

This may explain the findings of a low congruence between commercially available tumor next-generation sequencing (e.g., Foundation) versus liquid biopsies (e.g., Guardant360) in various malignancies [51,52,53,54]. According to Hahn et al. [55], the mean concordance rate was 17.6% (range 8.6% to 22.0%) between the two assays.

It is plausible that epigenetic markers are more useful than genetic markers for the purposes of separating CSCs from non-CSCs. It is conceivable that assays designed to detect circulating tumor cells (CTCs) and circulating tumor DNA (ctDNA)/cell-free DNA (cfDNA) by their epigenomic rather than genomic characteristics could be used to monitor and measure the elusive CSCs for diagnostic, prognostic, and predictive purposes by virtue of their biological and clinical relevance.

For example, Mo et al. [56] demonstrated that changes in ctDNA may enable us to detect residual disease, guide chemotherapy, stratify risk, and monitor recurrence of stage I to III colorectal cancer. They performed multiplex ctDNA methylation and quantitative polymerase chain reaction assay to detect the ctDNA of six biomarkers (SEPTIN9 locus 1, *BCAT1*, *IKZF1*, SEPTIN9 locus 2, *VAV3*, and *BCAN*)*,* which in one way or another implicated stem-like or stemness properties, i.e., pluripotency, self-renewal, and differentiation of progenitor stem cells in various tissues [57,58,59,60,61,62].

Taylor et al. [63] used digital bead-based ELISA technology to develop multi-cancer early detection assays. They measured LINE-1 (L-1) ORF1p in the plasma (at femtomolar levels) to detect minimal residual cancers and determine therapeutic response in various malignancies. Interestingly, epigenetic regulation (e.g., DNA methylation, histone modifications) strictly limit L-1 expression and function in somatic cells [64]. However, L1 mobilization tends to be prevalent but needs to be tightly controlled in pluripotent cells [64]. Otherwise, L-1 expression has the potential to disrupt genomic stability and instigate carcinogenesis.

One cannot help but notice that CSCs mimic and mirror normal stem cells (NSCs). CSCs mobilize and migrate like NSCs do. In contrast, non-CSCs (within solid tissues) tend to be moored, tethered, or leashed, like normal differentiated cells do (except for those in liquid media, such as neutrophils and erythrocytes). Therefore, CTCs are more like CSCs than non-CSCs, and ctDNA/cfDNA measures CSCs rather than non-CSCs. One predicts that plasma CTC numbers and ctDNA/cfDNA/proteome levels should be higher in advanced refractory than localized early-stage cancers.

Importantly, measuring plasma ctDNA/cfDNA/proteome levels may be particularly useful in a maintenance setting, when we are dealing with minimal residual disease after induction therapy in which the bulk of non-CSCs has been eradicated and only CSCs remain. When we need to measure relatively rare tumor entities like CSCs, there is a need for sufficient sensitivity and specificity that perhaps only pertinent stemness or stem-like ctDNA/cfDNA/proteome can provide.

### 5.2. Imaging

Biomarkers that brand CSCs or stamp non-CSCs may be useful for the purposes of identification and detection when it concerns imaging. For example, most conventional imaging is designed to detect the bulk of non-CSCs. When non-CSCs are eliminated and we are left with CSCs, it behooves us to discern what is left that still needs to be treated, i.e., CSCs rather than non-CSCs, in the residual tumor or progressive cancer.

Glucose vs. glutamate. In vivo imaging using PET and MRI is dependent in large part on glucose uptake and metabolism [65]. Therefore, when most active tumors comprise non-CSCs which utilize glucose, these scans are relevant and useful for our purposes and practices. However, when we become encumbered with refractory tumors that are comprised mostly of CSCs, in which glutamate rather than glucose uptake and metabolism may be pivotal if not predominant, in vivo imaging using PET and MRI based on glutamate rather than glucose may be more pertinent and informative. Hence, Seo et al. [66] used MRI to demonstrate that glutamine uptake is positively correlated with the distribution of the glutamine transporter, ASCT2, and certain CSC markers, such as CD44 and CD166, in a mouse xenograft model of HT29 human colorectal cancer cells.

Lactate and pyruvate. It is of interest that when c-myc (MYC) was turned on in tumors, multiple glycolysis-pathway genes were upregulated [67]. Other oncogenes, including activated alleles of RAS, AKT1, and PIK3CA, or loss of tumor suppressors (such as p53 and VHL) also increased glycolysis.

Lactate dehydrogenase A (LDHA) is a key regulator of glycolysis. LDHA abundance and activity in tumors are tightly correlated with in vivo pyruvate conversion to lactate. Conversion of pyruvate to alanine predominates in pre-cancerous tissues prior to observable morphologic or histological changes, and correlates with MYC activity. The global metabolic changes may be related to glycolytic activity, which is related to “stemness” metabolism that could be quantified based on high LDHA levels and visualized on ^13^C-pyruvate magnetic resonance spectroscopic imaging [68].

TFRC and PSMA. Because certain CSCs express transferrin receptor (TFRC) and prostate-specific membrane antigen (PSMA), TFRC and PSMA could be useful biomarkers for the directed imaging and targeted therapy of CSCs in cancer care.

Gjyrezi et al. showed [69] that TFRC expression increased with disease stage, from adenocarcinoma to CRPC and to the more aggressive neuroendocrine prostate cancer (NEPC). TFRC expression was also increased in EMT models and the AR-V7 variant of prostate cancer.

De Kouchkovsky et al. [70] showed that ^68^Ga-citrate (which avidly binds to circulating transferrin) PET could detect CRPC, presumably with MYC hyperactivity (since the TFRC is a direct MYC target gene) and neuroendocrine differentiation.

Interestingly, both TFRC and PSMA belong to the same transferrin receptor family of type II membrane glycoproteins [71]. As a CSC biomarker [72], PSMA is supposed to cross tissue-type barriers, i.e., it should not be prostate-specific. Indeed, Kasimir-Bauer et al. [73] has shown that PSMA is also a significant prognostic and predictive biomarker in TNBC.

Importantly, PSMA as a biomarker involved in CSC glutamate metabolism has already proven its value both as an imaging modality [74,75] and as a therapeutic option for the management of CRPC [76]. One would predict that TFRC as a biomarker involved in CSC iron metabolism may also reveal its clinical versatility and utility, provided we use it in the right setting at the right time.

### 5.3. Therapeutics

Because cancer is often a systemic rather than isolated problem and a dynamic rather than static process, we propose that anti-CSC treatments which control a whole CSC network rather than a single CSC signal pathway, such as those targeting CSC-related micro-RNA (miRNA) and histone deacetylase (HDAC), may be just as efficacious and safe as, and can be more cost-effective than, conventional salvage treatments when used at the right time in the right patients with the right cancer subtypes, and monitored with appropriate CSC biomarkers.

#### 5.3.1. MiRNA

Among the many epigenetic modifiers, miRNA is both versatile and specific. Humans have about 25,000 genes and more than 400 miRNAs. As many as 30% of all genes are thought to be under miRNA control. A miRNA regulates gene expression by base-pairing with target mRNAs and inhibiting their expression. Each miRNA can target many genes (dozens to hundreds). 

Utikal et al. [77] showed that a panel of miRNAs, which specifically activate or repress stemness traits such as pluripotency and self-renewal in NSC may also be operative in CSCs. Importantly, miRNA expression profiles may provide a reliable way to classify tumor subtypes by denoting or displaying the developmental lineages of their malignant cells of origin [78].

Furthermore, several miRNAs are exclusively expressed in progenitor stem cells but not in their progeny differentiated cells [77]. For example, the miR-34 family reduces the protein levels of important pluripotency-associated factors, such as NANOG, SOX2, and MYCN, and targets p53. Interestingly, miR34a is a powerful suppressor of carcinogenesis and metastasis. Similarly, the let-7 family negatively regulates the levels of pluripotency-associated proteins SALL4, MYC, and LIN28A, whereas miR-294 induces their expression. Not unexpectedly, certain specific circulating miRNAs have been found to be useful biomarkers for cancer care [79].

Importantly, diverse dietary bioactive components may exert their anti-tumor effects, in part through modulation of CSC miRNA expression [80,81,82]. Hence, curcumin (in turmeric) reduced EZH2 expression and increased a panel of tumor suppressive miRNAs, specifically let-7 family members, including miR-200b, and miR-200c [83]. Genistein (in soy) up-regulates miR-200, which was associated with down-regulation of validated targets ZEB1, slug (SNAI2), and vimentin (VIM), known to play a role in EMT [84]. Genistein also induced upregulation of miR-34a and mediated negative regulation of HOTAIR in PC3 and DU145 prostate cancer cells [85]. In addition, DIM (in cruciferous) up-regulated let-7b, let-7c, let-7d, let-7e, miR-200b, and miR-200c [84]. Epigallocatechin-3-gallate (EGCG, in green tea) notably upregulated miR-7-1, miR-34a, and miR-99a but reduced expression of miR-92, miR-93, and miR-106 [86]. Butyrate (in fermentable fiber/pectin) increases expression of miR-200c [87].

#### 5.3.2. HDACi

The epigenome provides us with a different perspective and narrative from that of the genome on the origin and nature of CSC vs. non-CSC. It also provides us an invaluable opportunity to investigate the molecular profiles of CSC (e.g., displaying EMT pathways, inflammatory factors, miRNA signatures, etc.) with diagnostic, prognostic, and therapeutic implications in cancer care.

Witt et al. [88] demonstrated that certain histone deacetylase inhibitors (HDACis) could eradicate breast and ovarian CSC. If CSC is a pertinent pathological entity, then eliminating it would be biologically meaningful and clinically impactful. Importantly, there are many natural HDACis found in food, e.g., diallyl disulfide (garlic and onions), sulforaphane-rich extracts (broccoli and other cruciferous, cabbage-family vegetables), polyphenols (green tea, turmeric, soy, berries) [89], that may modulate and control CSCs and keep them indolent, if not dormant, and can be taken for a prolonged period of time in a safe and affordable manner. In many respects, this is the essence of cancer chemoprevention.

For example, Pomi-T (comprising pomegranate, green tea, broccoli, and turmeric) compared with placebo provided a significant delay in median PSA rise for CSPC patients on active surveillance or watchful waiting following previous interventions [90]. Zyflamend (containing turmeric, holy basil, green tea, oregano, ginger, rosemary, Chinese goldthread, Hu Zhang, barberry, and basil skullcap) produced a PSA response (decrease by 25–50% from baseline) in 48% of patients with high-grade prostatic intra-epithelial neoplasia [91].

Specifically, curcumin (in turmeric) was one of only about 40 promising agents out of over 1000 tested by the NCI for chemoprevention activity [92]. Curcumin provides multi-faceted anticancer effects, promotes apoptosis, inhibits survival signals, scavenges reactive oxidative species, and reduces the inflammatory cancer microenvironment. Curcumin may enhance chemosensitivity [93] and radiosensitivity [94] by inhibiting abcb1b, as mediated by Pik3r1, Akt1, and Nfkb1 [95], by inhibiting ABCG2 [96], and by affecting CD44+ cells [97].

Curcumin decreased the side population, which is known to be associated with the stem cell population in the rat glioma cell line C6 [98]. A curcumin analog CDF decreased CSC markers (NANOG, POU5F1, EZH2, miR-21) in prostate cancer cells under hypoxic conditions [80]. Another curcumin analog, GO-Y030, inhibited the expression of STAT3 and suppressed CSC growth in colon cancer cells [99].

Importantly, when there are abundant and redundant angiogenic pathways and inflammatory factors, blocking one signaling pathway and targeting one oncogenic factor would be less rewarding and satisfying. However, by dampening the whole cancer network with anti-angiogenic agents (such as polyphenols) and quenching an entire tumorigenic cascade with anti-inflammatory products (such as turmeric), we predict that patients may experience prolonged remission (after response to the same standard approved treatments) and improved overall clinical outcome.

A word of caution and a caveat about CSC biomarkers—without a reliable surrogate biomarker that measures the presence or activity of CSCs we may not be able to determine any clinical benefit from treatments that target CSCs in a timely manner and abandon potentially salubrious treatments prematurely.

## 6. Clinical Implications

A proper cancer theory accounts for not only inter-tumoral heterogeneity but also for intra-tumoral heterogeneity, in which consideration of tumor location is paramount and timing of therapy is also a prerequisite for optimal patient care. In other words, we may need to treat and measure (by way of biomarkers) both the CSCs and non-CSCs in the primary tumor and metastatic disease in a neoadjuvant or adjuvant setting, depending on the type and nature of cancer. In addition, the origin and nature of cancer implicates different tumor subtypes with disparate stem cell derivations. Importantly, a stem cell origin of cancer advocates multimodal therapy over targeted therapy and integrated medicine over precision medicine. It promotes therapy development over drug development in cancer care.

Unfortunately, when we do not have a proper cancer theory, effective treatments against CSCs may be less evident or beneficial when used in an upfront setting (when CSCs comprise only a small fraction of the cancer) compared to a maintenance setting (when CSCs constitute the main and perhaps only components of the so-called minimal residual disease after induction therapy). In other words, treating CSCs may be invaluable in a preventive setting when they comprise the minimal nascent malignant cells before evolution and manifestation of the entire cancer, and in a maintenance setting when they constitute the minimal residual malignant cells after elimination of the bulk of non-CSCs by way of upfront or induction therapy.

Currently, many of our effective treatments have managed to control non-CSCs rather than CSCs. For example, androgen deprivation therapy (ADT), such as leuprolide and degarelix, and ARSIs, such as abiraterone and enzalutamide, have served us well for the treatment of the bulk of prostate non-CSCs that are dependent on AR signal pathways and can be detected in the tumor tissue and measured in the blood by the biomarker, prostatic specific antigen (PSA). However, these approved treatments do not cure CRPC, because they do not affect those residual castration resistant prostate CSCs, which do not depend on AR signal pathways and cannot be detected in the tumor tissue or measured in the blood by PSA.

Therefore, we predict that the conception and application of treatments which control or eliminate CSCs could provide substantive rather than nominal clinical benefits in cancer care [100,101,102,103]. How to complement novel CSC-targeted therapy with conventional non-CSC-targeted therapy is key in therapy versus drug development, to provide breakthrough rather than marginal improvements, and exponential rather than incremental improvements in clinical outcomes [11,104].

Indeed, knowledge of PSMA as a CSC biomarker suggests that anti-PSMA may also be anti-CSCs, which could change our current perspectives and narratives regarding drug development and render an even better treatment when integrated with our traditional anti-non-CSC drugs in our effort to optimize therapy development for patient care. Similarly, some of our heralded anti-cancer therapeutics, such as anti-PDL1 [105,106,107], anti-HER2 [35], anti-NECTIN4 [108,109], and anti-TROP2 [36,37], are also anti-CSC drugs that can be, or have already been, utilized in combination with anti-non-CSC treatments to maximize clinical outcomes.

## 7. Conclusions

The cancer biomarker plays an outsized role in cancer research and in cancer care. Often enough, its discovery is by design rather than by accident. A successful advancement and utilization of cancer biomarkers in cancer care depends in large part on our adoption and adherence to the scientific method and on our knowledge and understanding of the origin and nature of cancer.

If cancer is a genetic disease and has a genetic origin and nature, we search for genetic biomarkers. If cancer is a stem cell disease, we seek stemness biomarkers. A stem cell-related origin and nature of cancer suggest that genetic biomarkers may be pivotal, but cellular context is paramount.

When we consider cellular context and when it concerns separation of CSCs from non-CSCs, whether the biomarker is derived from DNA, RNA, or protein and whether it is detected within the tumor or in the blood (such as CTCs and ctDNA) has biological ramifications and clinical implications for cancer care.

Importantly, if CSC is a relevant biological and clinical entity, then monitoring, measuring, and modulating it by ways of its genome versus its epigenome become imperative for diagnosis, prognosis, and therapeutics in cancer care (Figure 1).

Otherwise, without the right knowledge of the origin and nature of cancer, we may make bad decisions based on good numbers and we may make incremental clinical progress for the wrong reasons. Unfortunately, when we do not formulate a proper cancer theory and when we do not adopt or adhere to the scientific method, we will learn that many promising biomarkers may lose biological value and lack clinical impact.

## Figures and Tables

**Figure 1 cancers-15-05533-f001:**
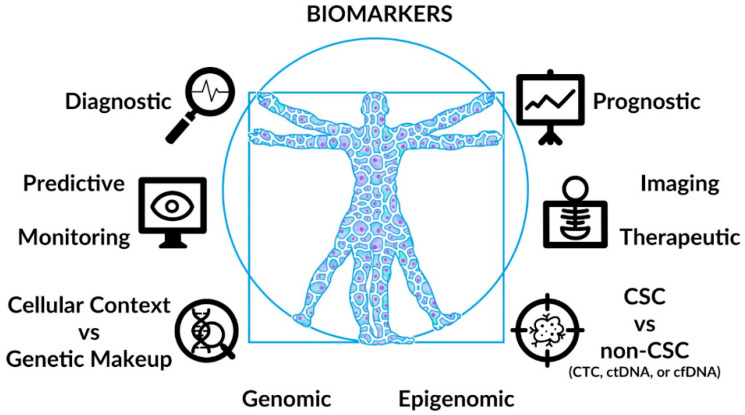
Biomarkers in cancer care as embodied by the Vitruvian virtues (stability, utility, and beauty) and principles (order, arrangement, eurythmy, symmetry, propriety, and economy). Reproduced with permission from Benjamin Tu (www.bentubox.com), accessed on 30 August 2023.

**Table 1 cancers-15-05533-t001:** Bladder cancer subtypes: biological phenotypes and therapeutic implications.

Subtypes (Incidence)	Biology	Treatment
Luminal-papillary (35%)	FGFR alterations	Low response to NAC
Luminal-infiltrated (19%)	EMT, PD-L1	CPI, resistance to NAC
Luminal (6%)		
Basal-squamous (35%)	PD-L1	CPI, NAC
Neuronal (5%)		NAC (EP)

NAC: neoadjuvant chemotherapy; CPI: checkpoint inhibitor; EP: etoposide and platinum.

**Table 2 cancers-15-05533-t002:** Renal cell carcinoma: stem cell origins and therapeutic implications.

ITH	SCNA	
LOW	HIGH	No surgery, may not respond to CPI or TKI
		Increased metastatic potential
HIGH		May benefit from surgery + CPI (PDL1+)
		Delayed metastatic potential
LOW	LOW	Benefit from surgery + TKI (angiogenesis+)
		Decreased metastatic potential

ITH: intra-tumoral heterogeneity; SCNA: somatic copy-number alteration; CPI: checkpoint inhibitor; TKI: tyrosine kinase inhibitor.

**Table 3 cancers-15-05533-t003:** Renal cell carcinoma subtypes: distinct clinical outcomes derived from tumors’ unique epigenome and genome.

ccrcc	1	2	3	4	
PD	22%	3%	0%	27%	
PR/CR	41%	53%	70%	21%	*p* = 0.005
PFS (mos)	13	19	24	8	*p* = 0.0003
OS (mos)	24	35	50	14	*p* = 0.0002
ccrcc4 and ccrcc1 = ITH LOW, SCNA HIGH: No surgery					
ccrcc3 = ITH LOW, SCNA LOW: Benefit from surgery					
ccrcc2 = ITH HIGH: benefit from surgery + TKI/CPI					

ccrcc: clear cell renal cell carcinoma; PD: progressive disease; PR/CR: partial response/complete response; PFS: progression-free survival; OS: overall survival.

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
