# Peer review of "Stem Cell Theory of Cancer: Clinical Implications of Epigenomic versus Genomic Biomarkers in Cancer Care"

_cancers, 2023, doi:10.3390/cancers15235533_

Round 1

Reviewer 1 Report

Comments and Suggestions for Authors

In the manuscript „Stem Cell Theory of Cancer: Implications of Epigenomic versus Genomic Biomarkers in Cancer Care” Tu and co-authors present some of their thoughts on the nature and origin of cancer, tumour heterogeneity, diagnosis and therapy.

I found the manuscript poorly organised, rambling and incoherent. Little clear and specific information is presented and when this is done, the factual evidence for it is sometimes rather questionable. For the most part, the manuscript indulges in general, abstract statements or diffuse ruminations about various loosely connected topics related to cancer, instead of actual data. Here is a sample, which, as far as I am able to understand, also illustrates the authors’ conflicted scepticism towards the scientific method: “Perhaps a pertinent theory is in the mind of the theoretician and a credible observation is in the eyes of the beholder. A potential problem with the scientific method is that it does not preclude us from performing exemplary experiments to test an erroneous theory based on faulty observations (such as those generated in the laboratory rather than in real life, in which the experiments were conducted to generate rather than to test the hypotheses). In the case of cancer biomarkers, when we do not adopt or adhere to the basic principles of the scientific method, what we discover are likely to be less useful, if not useless, for cancer care. When we happen to find some scientific merits and clinical values in some biomarkers this way, we may have done so for the wrong reasons.

Presumably, the “stem cell theory of cancer” is supposed to be the common thread of the review, but the link to this topic often gets lost. Overall, after reading this manuscript, I did not feel that I had gained much concrete and useful knowledge from it or any really novel and specific conceptual understanding.

The manuscript does not even contain a clear definition of cancer stem cells and an explanation of their role in tumour development and progression. The “stem cell theory of cancer” is contrasted with the “genetic theory of cancer”, but neither theory is properly explained, nor are the differences between them or the fact that they are not mutually exclusive.

The title of the manuscript suggests that it might present a systematic overview of cancer stem cell (CSC) biomarkers and their putative utility, but this is not the case. CSC markers are only mentioned sporadically and often without proper justification from the cited literature that they are in any way specific for CSCs. For example, BAP1 is postulated to be a CSC biomarker in ccRCC4, but I did not find evidence substantiating such a claim in the cited studies. Similarly, methylated ctDNA for biomarkers including SEPTIN9, BCAT1, IKZF1, VAV3, and BCAN is claimed to implicate “in one way or another [...] stem-like or stem-ness [sic] properties”, but this is not supported by the cited literature. Alanine is also proposed as a CSC biomarker in tumour imaging, again without supporting evidence from the cited reference.

In terms of therapy, anti-PDL1, anti-HER2 and others are presented as “anti-CSC drugs”, but there is no discussion of the fact that these are by no means specific for CSC (or even cancer cells in general). Other proposed interventions are even less specific, e.g. histone deacetylase inhibitors or dietary components, which might modulate the expression of miRNAs that have been linked, among many other things, to CSCs.

The manuscript carries an undercurrent of disparagement for “the fallacy of precision medicine”, arguing for promoting “therapy development over drug development in cancer care”. The section on (allegedly CSC-targeting) therapeutics mostly focuses on the use of dietary components and herbal supplements, such as curcumin, polyphenols and “Pomi-T”. To say the least, the specificity of any of those for CSCs is highly questionable.

In the introduction several sentences are directly copied from the abstract. Furthermore, self-citation is rather profuse. Interestingly, at least three of these self-citations, with the same first author, have a title that also begins with “Stem Cell Theory of Cancer:”, and two of them have already been published in Cancers over the last couple of years.

Reviewer 2 Report

Comments and Suggestions for Authors

Discuss the role of CSC biomarkers in primary and mixed tumors. Provide examples and evidence of their significance in different tumor types and stages.

Describe in detail the specific biomarkers that can identify cancer stem cells (CSC) and differentiate them from non-CSC. Explain their relevance in elucidating intertumoral and intra-tumoral heterogeneity and their potential impact on current prognostic and predictive tests.

The authors need to clarify the importance of cancer biomarkers in cancer research and care. Provide specific examples of how biomarkers have contributed to advancements in diagnosis, prognosis, and therapeutics.

 It's important to highlight the deliberate design and systematic approach used to identify relevant biomarkers.

Provide a thorough analysis of the different types of biomarkers derived from DNA, RNA, and protein. Explain their respective strengths and limitations and discuss their clinical implications.

Address the significance of formulating a proper cancer theory and adhering to the scientific method in biomarker research. Discuss the implications of not following these principles on the biological value and clinical impact of biomarkers.

Reviewer 3 Report

Comments and Suggestions for Authors

The manuscript title is interesting however the content and flow of the review should be further improved.

The authors provided a very short section on “brief history” which dated as early as 1847 to biomarker for multiple myeloma. It appears 1847 was the first ever time a biomarker was described for cancer? Is this true? I am not able to comment/ agree on this fact and the information provided since I am not a historian and not able to access to such old manuscripts personally. Should the editors accept this manuscript, please allow a more senior person to review this section or have the authors provide the full manuscript for the reviewers to review these facts especially for references from No 1-5.

It is not clear why the authors chose to discuss selected tumour types under the section primary and section mixed tumours. These seem too brief. Why only breast, lung and testicular tumors are briefly mentioned, let alone discussed. Other tumours that can have mixed histological variants include bone neoplasm like osteosarcoma was not even discussed.

Section 5.5 I believe this is an error? Why would 5.5 be there when the sentence was supposed to describe the ccrcc? Please write the review on ccrcc in sentence and not list such. In fact, the whole list from “favourable outcome…. Until…. inflamed” can be omitted and just describe the findings in the text as how it was done.

Section 6.0 – only has three subsections on imaging, monitoring, and therapeutics. However, when referring to the Diagram the authors provided in Figure 1, should also include predictive, prognostic and diagnostic markers. The review seemed to have no proper flow and many unnecessary points related to the subject / title of this review was included such as sections 1, 2 and 3 was not necessary but the review should focus on the evidence and roles of cancer stem cells in cancer epigenomics and genomics. Perhaps all the sections can be re-organized to address the subject more neatly.

Reference listed – error in the year. It should be 2019 instead of 2029. (Levine AJ, Jenkins NA, Copeland NG. The roles of initiating truncal mutations in human cancers: The order of mutations and tumor cell type matters. Cancer Cell 2029; 35:10-15.)

Comments on the Quality of English Language

Language editing can be considered if the manuscript has been revised. 

Round 2

Reviewer 1 Report

Comments and Suggestions for Authors

I thank the authors for their response to my comments and the modifications they have introduced in the text.

Unfortunately, I do not think that the explanations provided by the authors in the rebuttal letter or the changes in the manuscript address my concerns from the first round of revision to an appreciable extent.

In response to the criticism that many of the claimed CSC biomarkers do not have clear basis in the literature, the authors have cited work showing that some of the molecules have functions in embryonic stem cells, induced pluripotent stem cells and other non-cancer systems. This is not evidence that they are useful CSC biomarkers in cancer care.

The authors may be right that for some cancer patients an unspecific therapy, e.g. chemotherapy, HDAC inhibitors or herbal supplements, could have a more beneficial effect (or fewer adverse effects), compared to a drug that aims to target CSCs specifically. However, I don’t see how this supports the importance of a stem cell theory of cancer or informs us in any way about CSC biomarkers.

Reviewer 2 Report

Comments and Suggestions for Authors

Acceptable revision 

Reviewer 3 Report

Comments and Suggestions for Authors

The manuscript has been revised by the authors to improve the content however the flow / organization of the content is still not up to satisfaction. Having said that the manuscript is acceptable for publication but the impact to the society working on epigenomics and genomics of CSC is not certain.

Comments on the Quality of English Language

Have the manuscript edited by a native English speaker.